# In-House 3D-Printed vs. Conventional Bracket: An In Vitro Comparative Analysis of Real and Nominal Bracket Slot Heights

Luca Brucculeri [1], Claudia Carpanese [2], Mario Palone [3] and Luca Lombardo [1],*

1 Postgraduate School of Orthodontics, University of Ferrara, 44121 Ferrara, Italy
2 Private Practice, 44121 Ferrara, Italy
3 Department of Orthodontics, University of Ferrara, 44121 Ferrara, Italy
* Correspondence: dott.lulombardo@gmail.com

**Abstract: Aims:** The purpose of this study was to evaluate the accuracy of the slot height of in-house 3D-printed resin brackets, comparing them with other types of brackets on the market today, both ceramic and metallic. **Methods:** Seven different types of bracket systems were selected. For each system, ten brackets for tooth 2.1 with $0.022 \times 0.028$-inch slots were selected (total n° 70). Considering the whole sample, five types were commercially available and two were in-house 3D-printed. The entire sample was divided into four different groups according to the bracket material and the method of holding the archwire. Precision pin gauges with 0.002-mm increments were inserted inside the slot of each bracket, and the slot heights were measured, microscopically ensuring that the gauge completely filled the slot, with full contact between both the bottom and the top of the slot. **Results:** With respect to the other five types of brackets on the market, the two types of in-house 3D-printed resin brackets showed great accuracy of slot height ($0.558 \pm 0.001$ mm). There was a statistically significant difference between the real height measured and the nominal height declared by the manufacturers ($p < 0.05$) of all the samples investigated, with the exception of in-house 3D-printed resin brackets. Furthermore, the difference in slot height accuracy between commercially manufactured and in-house 3D-printed resin brackets was statistically significant. **Conclusions:** In-house 3D-printed resin brackets have a remarkably precise slot height, unlike commercially available brackets, whose slot heights tend to be significantly oversized with respect to the nominal values declared by the manufacturers.

**Keywords:** custom-made appliance; digital orthodontics; CAD–CAM





## 1. Introduction

Three-dimensional printing is being adopted at an increasing rate in various fields of dentistry, among them being orthodontics [1]. Currently, the most commonly used technologies are Polyjet and stereolithography [2]. Digital light processing (DLP) is a subset of the latter that relies on the projection of a flat image into light-curable resin [3–6] via an image-projection method developed by Texas Instruments (Dallas, TX, USA) in the 1980s. It uses a series of chipsets that exploit micro-electromechanical optical technology to process working light sources into photosensitive materials. The main functional part of the system is a digital micromirror device (DMD), which consists of a group of controllable mirrors of micrometric sze [7].

Patients' growing demand for orthodontic treatment that is as aesthetic as possible has resulted in the increasingly widespread use of aesthetic brackets as an alternative to traditional metal brackets with superior optical properties [8]. Another option for aesthetic treatment is the lingual technique [9], which, despite being a proven and successful method, is affected by several disadvantages, including difficult clinical management, reduced comfort, increased treatment times, especially in extraction cases, and the problematic resolution of severe rotations [9].

Vestibular fixed orthodontics with ceramic brackets are therefore often a preferred option. However, despite the superior optical properties of ceramic brackets, they have several mechanical disadvantages compared to traditional metal ones [10]. In particular, ceramics being a fragile material means that ceramic brackets are more prone to fracture (for example at the level of the bracket fins) upon mastication or the application of torque bends on the archwire. In addition, having a low resistance to tension, they do not have the ability to deform plastically, and debonding is therefore associated with an increased risk of enamel damage [10–13].

The brackets currently on the market, both metal and ceramic, are produced through procedures such as casting, injection moulding, milling and sintering [11,14–16]. The choice of material and the manufacturing technique influence both the final quality of brackets and their degree of precision in transmitting orthodontic forces, especially second- and third-order information [17,18]. Lefebvre et al., in a study carried out in 2019, demonstrated that the manufacturing techniques of brackets currently on the market do not offer an adequate degree of precision, with as many as 90–97% of brackets presenting slot height inaccuracies with respect to the nominal values declared by the manufacturers [17]. Although slot height is one of the fundamental aspects influencing third-order archwire–slot play and torque expression [18], according to the literature, the slot heights of brackets on the market, both ceramic and metal, tend to be oversized with respect to the nominal values declared by the manufacturers [19–21]. In fact, Arreghini et al. showed that in reality, bracket slots, both vestibular and labial, tend to be oversized by a percentage of between +0.56% and +11.6% [18].

Dimensional imprecision in orthodontic brackets, especially slot height, can affect the archwire–slot play [18,22–26]. In fact, Meling et al. have shown that even a slight increase in slot height leads to a considerable increase in the archwire–slot play, reducing the torque exerted on the teeth [22]. An adequate expression of torque is particularly important in extraction cases, as maintaining the torque on the anterior elements is necessary to allow adequate space closure [9]. Torque control is also essential for vestibulolingual root movements, as in the case of the recovery of ectopic teeth [9].

As an alternative to ceramic aesthetic brackets, computer-aided design/computer-aided manufacturing (CAD/CAM) technology has made it possible to produce biocompatible resin brackets. Considering the precision of the printing process now achieved through additive technology by modern 3D printers for dental use [19], the latter may be superior to the former in many respects. In fact, the increasing emphasis on the research and development of high-performance biocompatible resins has made it possible to produce aesthetic brackets with more favourable physical properties than those of ceramic aesthetic brackets and, in some cases, comparable to metal ones [19]. Furthermore, the CAD/CAM production of aesthetic brackets is practical, with brackets being printed as needed and customised according to clinical needs. Moreover, their production is economical for both the clinician and the patient.

While the production of in-house 3D-printed (IH3D) resin brackets via CAD/CAM technology offers numerous advantages for both the clinician and the patient, the accuracy of their bracket slot height reproduction remains to be verified through scientific studies. Considering the importance of the slot for the expression of torque, the purpose of this study was therefore to investigate the slot height accuracy of in-house 3D-printed resin brackets, as compared to other types of metal traditional brackets, self-ligating brackets, and aesthetics brackets on the market. As the data in the literature suggests that actual slot height tends to be oversized with respect to the nominal height [18], the null hypothesis was that there would be no difference in this regard between commercially available brackets and those 3D-printed in-house, i.e., that the CAM moulding process does not faithfully reproduce the CAD-phase design.

## 2. Material and Methods

The protocol for this in vitro study (number 2/2022) was approved by the Postgraduate School of Orthodontics Ethics Committee.

### 2.1. Sample Selection

Ten of each of seven types of vestibular brackets of size 0.022 × 0.028-inch for tooth 2.1 were investigated, making a total of 70 brackets. Specifically, the slot heights of five types of commercially available brackets were measured, each bracket from a different batch, while two types were 3D-printed in-house (Table 1). The sample was divided into four different groups according to the bracket material and the method of holding the archwire, as follows:

Group 1. <u>Conventional metal brackets</u>: Primo SWM (Sweden and Martina, Italy) and Legend Mini (GCOrthodontics, Breckerfeld, Germany).

Group 2. <u>Self-ligating metal brackets</u>: SlX 3D careers (Target Ortodonzia srl, Garbagnate Milanese, Italy) and Damon (Ormco, Glendora, CA, USA).

Group 3. <u>In-house 3D-printed resin brackets (IH3DB)</u>: IH3DB1, manufactured using SprintRay Pro 95 (SprintRay Inc., Los Angeles, CA, USA), and IH3DB2, manufactured using EnvionTec D4K (EnvisionTec, Gladbeck, Germany).

**Table 1.** Description of the sample for each type of bracket investigated.

| Brackets Selected for the Study | | | | |
|---|---|---|---|---|
| **Type of Bracket** | **Manufacturer** | **Tooth** | **Slot Height** | **Torque** |
| Primo SWM | Sweden & Martina | 2.1 | 0.022-inch/0.558 mm | 17° |
| Legend Mini | GC Orthodontics | 2.1 | 0.022-inch/0.558 mm | 17° |
| Damon Q2 | Ormco | 2.1 | 0.022-inch/0.558 mm | 12° |
| Carriere SLX 3D | Carriere | 2.1 | 0.022-inch/0.558 mm | 17° |
| Inspire ICE | Ormco | 2.1 | 0.022-inch/0.558 mm | 17° |
| IH3DB 1 | In-office (Sprintray Pro 95) | 2.1 | 0.022-inch/0.558 mm | 17° |
| IH3DB 2 | In-office (Envisiontec D4K) | 2.1 | 0.022-inch/0.558 mm | 17° |

The in-house 3D-printed resin brackets were digitally designed (CAD) using the professional 3D drawing software Rhinoceros 7 (Robert McNeel and Associates, Washington, WD, USA), and 3D printed (CAM) via two different desktop 3D printers using DLP (Digital Light Projection) technology. In brief, once the bracket had been designed, the .stl file was imported into dedicated 3D printing software, namely RayWare 2.8.1 (SprintRay Inc., Los Angeles, CA, USA), for the SprintRay Pro 95 printer (SprintRay Inc., Los Angeles, CA, USA) and the Envison One RP software (EnvisionTec, Gladbeck, Germany) for the EnvionTec D4K (EnvisionTec, Gladbeck, Germany).

For both printers, the various samples and supports were digitally designed and positioned on the printing plate as per Figure 1. The resins used to print the brackets were the following:

- Verseo Smile Temp (BEGO GmbH and Co. KG, Bremen, Germany) biocompatible resin, certified IIa with flexural strength ±80 Mpa and colour A2 on the Vita scale, calibrated for the SprintRay Pro 95 printer at 50 μm on the Z axis.
- C&B MFH (NextDent B.V. Soesterberg, Netherlands) biocompatible resin, certified IIa, with flexural strength ±107 Mpa, colour A2 on the Vita scale, and calibrated for the EnvionTec D4K printer at 100 μm on the Z axis.

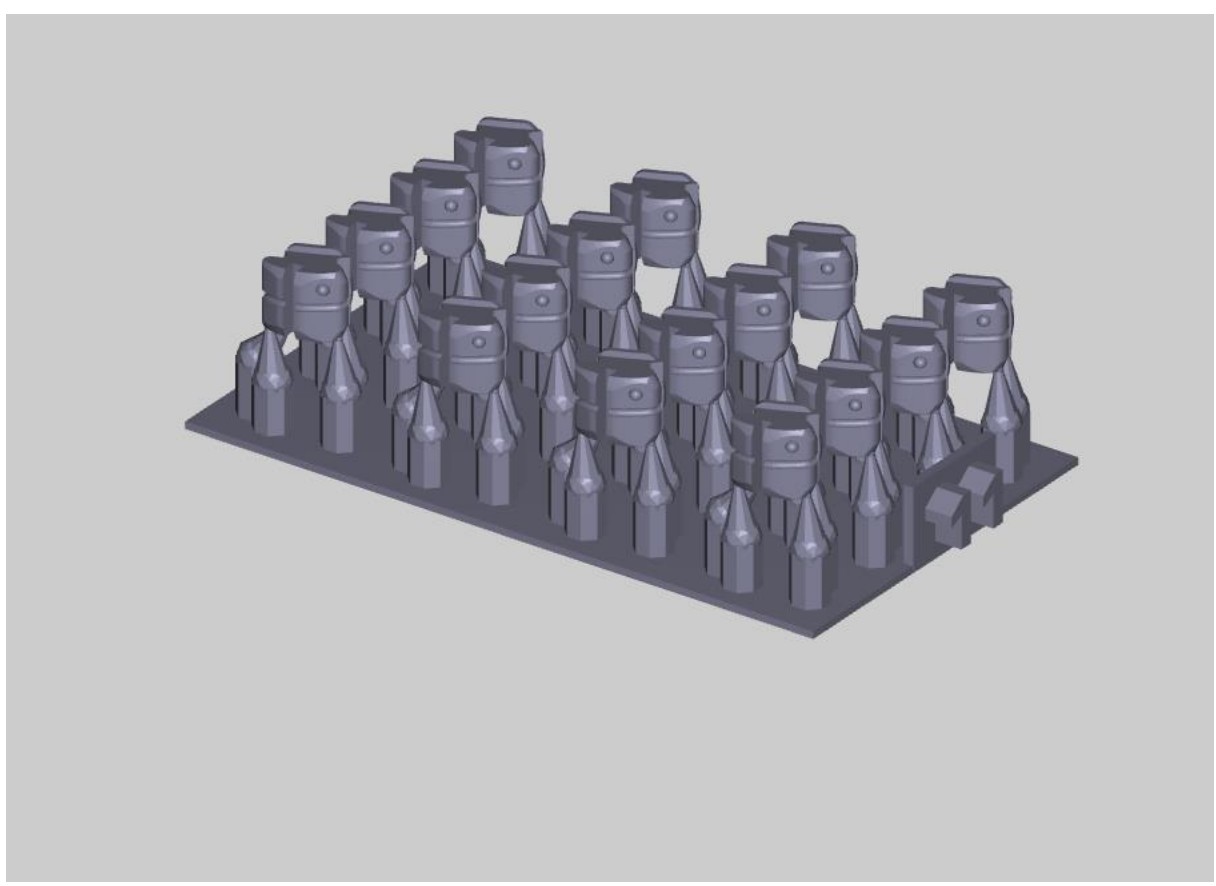

**Figure 1.** Stl files with digitally designed supports.

Azurea pin gauges (Azuréa Technologies S.A., Switzerland) with increments of 0.002 mm were used to measure the height of each slot (Figure 2). The gauges were inserted inside the slot of each bracket, beginning with the gauge with the greatest diameter (0.584 mm), and an Optika B500 optical microscope (Optika S.R.L, Ponteranica, Italy) was used to check that the gauge was in full contact with the bottom of the slot (Figure 2). The slot height was recorded as the diameter of the gauge that filled the slot completely from top to bottom (Figure 2). Measurements were conducted by a single operator (CC) and repeated two weeks apart.

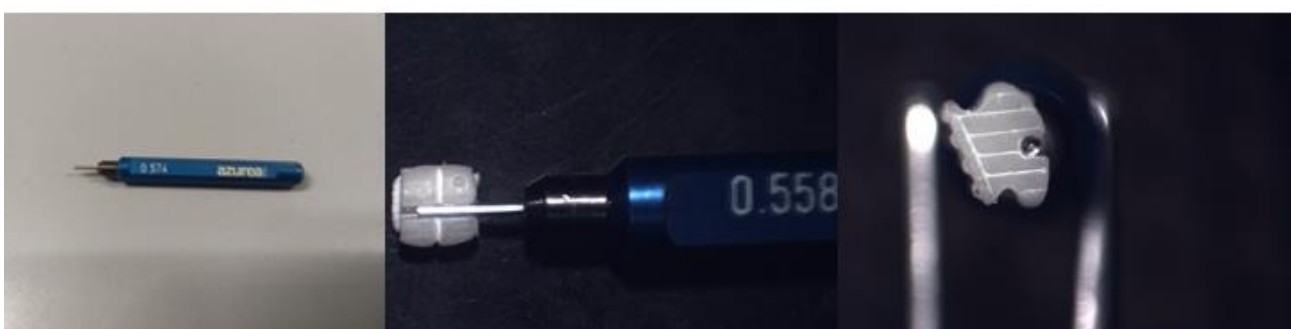

**Figure 2.** Azurea pin gauges (Azuréa Technologies S.A., Switzerland) and the microscopic image of the gauge inserted in the bottom of the slot.

### 2.2. Statistical Analysis

Microsoft Excel tables (Microsoft, Washington, DC, USA) were used to record the data collected, which were analysed using IBM SPSS v28 software (IBM, Endicott, New York,

NY, USA). Descriptive statistics were produced for each group, reporting the respective mean and standard deviation (SD).

Subsequently, for each of the seven types of bracket, the measured slot height was compared with the nominal slot height declared by the manufacturers (0.558 mm) to verify the production accuracy. The Wilcoxon nonparametric test ($p < 0.05$) was used for this purpose.

The Kruskal–Wallis test ($p < 0.05$) was performed on the entire sample in order to highlight any significant differences between measured and nominal height in each group. In the event of significant differences, the individual groups were compared with each other using the Dunn–Bonferroni test ($p < 0.05$).

Finally, within Groups 1, 2, and 4, each characterised by the presence of two types of brackets, the degrees of slot height accuracy of each brand were compared using Wilcoxon's non-parametric test ($p < 0.05$) in order to reveal any differences attributable to the different production methods used.

The non-parametric Wilcoxon test was also used to assess the reproducibility of the measurements ($p < 0.05$ considered significant). The reproducibility of each manual measurement performed was good, as the *p*-value was always >0.05 (Table 2) [20].

**Table 2.** Analysis of the repeatability of measurements by Wilcoxon's non-parametric test ($p < 0.05$).

| Type of Bracket | *p*-Value |
|---|---|
| Primo SWM | 0.18 |
| Legend Mini | 0.157 |
| Damon Q2 | 0.157 |
| Carriere SLX 3D | 0.157 |
| Inspire ICE | 0.157 |
| IH3DB 1 | 0.157 |
| IH3DB 2 | 1.0 |

## 3. Results

A descriptive analysis of each type of bracket investigated is reported in Table 3. Each type comprised a sample of 10 brackets (n = 10). The range of measurements made was expressed as the mean and SD range, from a minimum of 0.558 ± 0.001 mm (IH3DB2) and 0.558 ± 0.002 mm (IH3DB2) up to a maximum of 0.0578 ± 0.004 mm (Legend Mini and Inspire Ice brackets) and 0.0578 ± 0.002 mm (Damon Q2 self-ligating). Considering the minimum and maximum variation in the measured height as a percentage of the nominal height, there was a range from −0.36% (IH3DB1 and IH3DB2) to 4.30% (Primo, Legend Mini, Carriere SLX 3D and Inspire Ice) (Table 3).

Considering each individual type of bracket, there was a statistically significant difference between the real, measured slot height and the nominal, declared slot height ($p < 0.05$) for all the bracket samples investigated, with the exception of IH3DB1 ($p = 1$) and IH3DB2 ($p = 0.317$) (Table 3). In other words, the 3D-printed brackets presented the greatest accuracy in terms of slot height. Comparing the four different groups of brackets (Group 1: metal, conventional; Group 2: metal, self-ligating; Group 3: ceramic; and Group 4: IH3D), the Kruskal–Wallis test revealed statistically significant differences in slot height accuracy (H3 = 43.994, $p < 0.001$, n = 70) (Figure 3). Subsequent pairwise comparisons showed significant differences between the following pairings: Group 1 vs. Group 4 ($p < 0.01$), Group 2 vs. Group 4 ($p < 0.01$), and finally, Group 3 vs. Group 4 ($p < 0.01$) (Table 4).

**Table 3.** Descriptive analysis of each type of bracket investigated. SD: standard deviation; CL: confidence limit; SE: standard error and comparative analysis of measured and nominal slot heights for each type of bracket investigated ($p < 0.05$ *).

| | Type of Bracket | n | Mean (mm) | DS (mm) | Minimun Value (mm) | Maximum Value (mm) | CL (95%) (mm) | SE (mm) | Minimun Percentage Variation (%) | Maximum Percentage Variation (%) | *p*-Value |
|---|---|---|---|---|---|---|---|---|---|---|---|
| **Group 1** | Primo SWM | 10 | 0.571 | 0.007 | 0.564 | 0.582 | 0.543–0.601 | 0.0021 | 1.08% | 4.30% | 0.005 * |
| | Legend Mini | 10 | 0.578 | 0.004 | 0.572 | 0.582 | 0.549–0.607 | 0.0011 | 2.51% | 4.30% | 0.004 * |
| **Group 2** | Damon Q2 | 10 | 0.578 | 0.002 | 0.574 | 0.580 | 0.549–0.607 | 0.0007 | 2.87% | 3.94% | 0.005 * |
| | Carriere SLX 3D | 10 | 0.576 | 0.005 | 0.566 | 0.582 | 0.547–0.605 | 0.0016 | 1.43% | 4.30% | 0.005 * |
| **Group 3** | Inspire ICE | 10 | 0.578 | 0.004 | 0.572 | 0.582 | 0.549–0.607 | 0.0011 | 2.51% | 4.30% | 0.005 * |
| **Group 4** | IH3DB 1 | 10 | 0.558 | 0.002 | 0.556 | 0.560 | 0.530–0.586 | 0.0005 | −0.36% | 0.36% | 1 |
| | IH3DB 2 | 10 | 0.558 | 0.001 | 0.556 | 0.560 | 0.530–0.586 | 0.0004 | −0.36% | 0.36% | 0.317 |

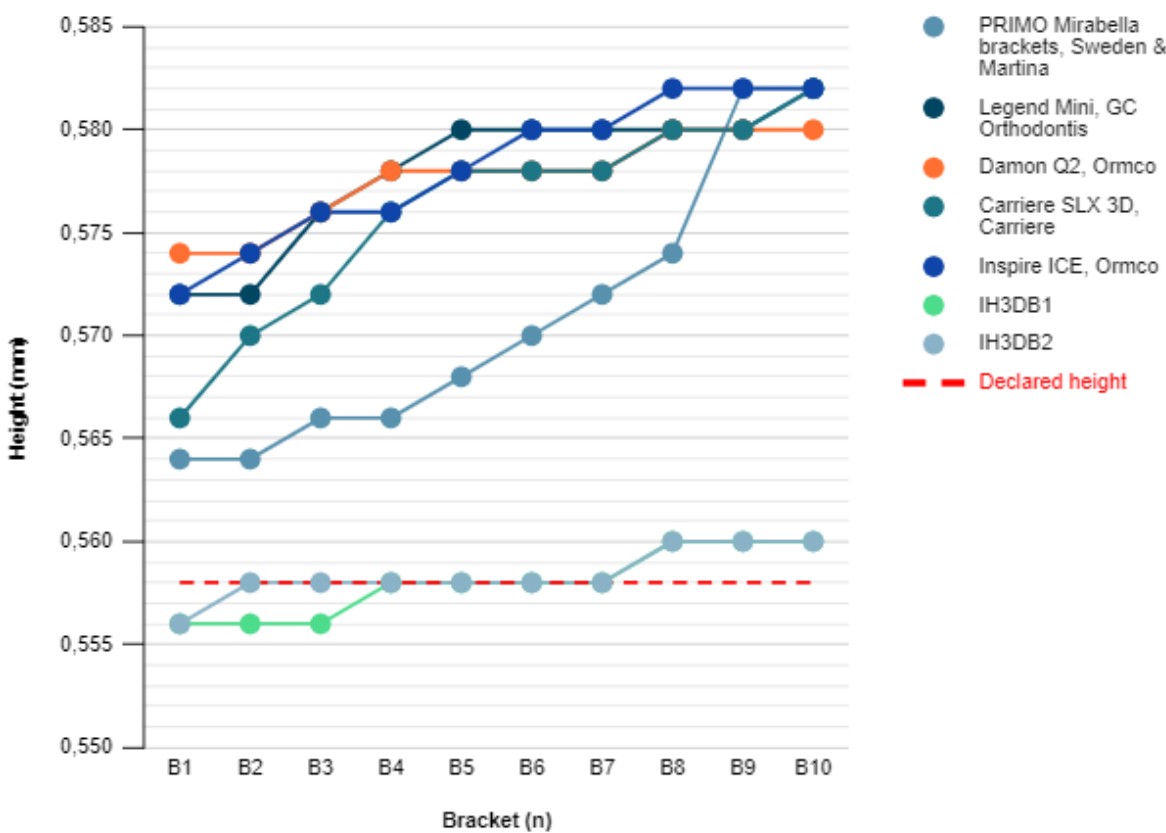

**Figure 3.** Accuracy of slot height in the various groups.

**Table 4.** Pairwise Dunn–Bonferroni comparative analysis of slot height in the four groups investigated ($p < 0.05$ *).

| Groups | Test Statistic | SE | Std. Test Statistic | *p*-Value |
|---|---|---|---|---|
| Group 3 vs. Group 2 | 5.1 | 7.832 | 0.651 | 0.515 |
| Group 3 vs. Group 1 | 8.525 | 7.832 | 1.089 | 0.276 |
| Group 3 vs. Group 4 | −40.45 | 7.832 | −5.165 | <0.001 * |
| Group 2 vs. Group 1 | 3.425 | 6.394 | 0.536 | 0.592 |
| Group 2 vs. Group 4 | −35.35 | 6.394 | −5.528 | <0.001 * |
| Group 1 vs. Group 4 | −31.925 | 6.394 | −4.993 | <0.001 * |

Finally, the slot height accuracy of brackets belonging to the same group (Groups 1, 2, and 4) was compared. This highlighted statistically significant differences in Group 1, the conventional metal brackets ($p < 0.001$) (Table 5).

**Table 5.** Wilcoxon's within-group comparative analysis of the bracket types 1, 2, and 4 ($p < 0.05$ *).

| Groups | Typology of Bracket | | *p*-Value |
|---|---|---|---|
| Group 1 | PRIMO SWM | Legend Mini | 0.010 * |
| Group 2 | Damon Q2 | Carriere SLX 3D | 0.102 |
| Group 4 | IH3DB 1 | IH3DB 2 | 0.157 |

## 4. Discussion

The accuracy of slot height reproduction is one of the many aspects of brackets that must be thoroughly investigated if their reliability is to be assured and their routine use endorsed. This study demonstrated that, across a sample comprising metal, ceramic, and resin brackets, the percentage difference between the real and nominal slot heights ranges from −0.36% to +4.30%. Specifically, the slots of all brackets investigated, except the 3D-printed ones, were significantly oversized with respect to the nominal values declared by the manufacturers (from +1.08% to +4.30%). This rejects our null hypothesis, i.e., that there would be no difference in this regard between commercially available brackets and those 3D-printed in-house

Although an increase of +4.30% may seem small, the application of Meling et al.'s formula highlights how this leads to considerable increases, about 19°, in the archwire–slot play [22].

The results of this study are in agreement with those presented in the literature [18,22–26], although smaller than the dimensional variation found by other authors. For instance, Cash et al. reported that slot height tends to be oversized by between +5% and +24% [25]. Similarly, Arreghini et al. found that lingual and vestibular bracket slot heights are between +0.56% and +11.6% oversized [18], a conclusion shared by Joch et al., who reported vertical oversizing ranging from 1% to 7% as compared with the nominal values [24].

The manufacture of metal brackets mainly relies on metal injection moulding (MIM) and milling, while ceramic brackets are usually made via ceramic injection moulding (CIM) [27,28]. The literature reveals differences in precision both among and within brands and suggests that inaccuracies can be introduced at different stages during the process of manufacturing metal brackets. Inaccuracies may be introduced during slot milling, which can be affected by the size and extent of the vibration of the cutter and the subsequent stages of finishing and polishing. The precise reproduction of injection-moulded brackets, on the other hand, will depend on the accuracy of the mould. Like the metal brackets on the market, ceramic bracket slots also tend to be oversized. As with the MIM process, the CIM process also has the same production steps at which errors can accumulate [28], and firing may also be responsible for altering slot size and shape.

In fact, Cash et al. [25], who measured the slot heights of 0.022-inch metal brackets from six different manufacturers, found a range of inaccuracy of 5–24%, with four types of brackets showing parallel slot walls, five having converging slot walls, and two with divergent slots. They also noted that some samples from the same manufacturers had convergent and others divergent walls, resulting in a variation in height between the slot entrance and exit. Similarly, Brown et al. compared 100 different brackets from 10 different manufacturers, concluding that manufacturing anomalies can affect both individual brackets and entire series of brackets [28].

This disparity was reflected in Group 1 (conventional metal brackets) of this study, in which a direct comparison between the conventional metal brackets Primo and Legend Mini revealed different degrees of inaccuracy, whereas Group 2 brackets (self-ligating metal) were more homogeneous.

While the direct comparison of metal and ceramic brackets (Groups 1, 2 and 3) revealed similar levels of inaccuracy, with no statistically significant differences, a comparison between groups revealed that in-house 3D-printed resin brackets (Group 4) presented significantly greater slot height accuracy as compared to conventional brackets. A curious aspect of the study presented here is that the IH3DB slots were, in some cases, slightly undersized, with negative percentage differences from nominal values ($-0.36\%$). This aspect, never yet reported in the literature, seems to depend on contraction during the post-curing phase of the 3D printing process [29]. In fact, contraction rates of 6–10% have been reported by Cervera et al. [29], likely arising from thermal expansion and contraction phenomena during bracket production [30–33]. Several other factors must be taken into account in order to limit this phenomenon, including exposure time, wavelength and power supply, as well as differences in filler particles and polymerisation methods [34–36]. That being said, Moon et al. have shown that DLP technology printers are associated with fewer contraction phenomena than the others [28], despite a degree of contraction on both the X and Y axes, as compared to the Z axis [7], making the position of the object to be printed on the print plate fundamental.

Despite the inherent limitations in the 3D-printing process, the findings from this study are encouraging. In fact, the analysis performed revealed that although there was a significant difference between the nominal and measured slot heights of the five types of commercially available brackets (conventional and self-ligating metal and ceramic) tested ($p < 0.05$), the same could not be said for the 3D-printed brackets. Indeed, the slot heights measured for both IH3DB1 and IH3DB2 brackets were very similar to those included in the CAD design. Moreover, this reproduction accuracy was reflected across the sample of IH3D brackets, as evidenced by the very low standard deviation. This compares favourably with the commercially available brackets, which all present oversized slots and inconsistent measurements within the same sample, and indicates that, with current technologies, 3D printing is a precise and reproducible means of manufacturing custom brackets.

As the 3D printing of resin brackets is of very recent application, the literature on the subject is still very scarce. However, the results obtained appear to be very encouraging, as no obvious differences were found when using a printer at 50 µm resolution or 100 µm on the Z axis with the models reported herein. Further studies should aim to investigate the printing accuracy of other professional printers used in dentistry. Other aspects of IH3D brackets should also be investigated, including wear resistance, the quantification of frictional forces generated with the archwire, the in vivo and in vitro testing of the colour stability, and any deformation following the application of torque bends.

*Limitations and Future Recommendations*

Despite being innovative, some limitations of the current study should be mentioned. First of all, a small number of brackets were measured for each group, and future investigations will aim to both augment the sampling extent and introduce other commercially available brackets.

Moreover, the shape of the slot of the IH3D resin brackets and its possible wearing under the application of torque movements was not yet investigated, although it could negatively affect torque expression.

According to these initial findings, a reduction in straight-wire appliance prescriptions should be considered, limiting the overcorrections usually applied in order to counterbalance the increase in the archwire–slot play present with the use of conventional brackets.

**5. Conclusions**

The null hypothesis was rejected, and the following conclusions can be drawn:

- The slots of all brackets investigated except the 3D-printed ones were significantly oversized with respect to the nominal values declared by the manufacturers (from +1.08% to +4.30%).
- In-house 3D printing provides brackets with remarkable slot height precision (from −0.36% to +0.36%) and consistency, although in some cases, the slots were slightly undersized, likely due to contraction phenomena during the post-curing phase.
- Three-dimensional printer variables and the type of resin used did not prove to be a factor affecting the accuracy of bracket slot height in this sample, as there were no statistically significant differences between IN3DB1 and IN3DB2.

**Author Contributions:** Performed the concept and design of the study and search strategy: L.B. and L.L. Performed the data extraction and qualitative synthesis and wrote the manuscript: C.C., L.B. and M.P. Revised the manuscript critically for important intellectual content: L.L. and M.P. Approval of the version of the manuscript to be published: L.L., M.P., L.B. and C.C. All authors have read and agreed to the published version of the manuscript.

**Funding:** This research received no external funding.

**Institutional Review Board Statement:** Not applicable.

**Informed Consent Statement:** Not applicable.

**Data Availability Statement:** Not applicable.

**Conflicts of Interest:** The authors declare no conflict of interest.

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
