# Peer review of "In-House 3D-Printed vs. Conventional Bracket: An In Vitro Comparative Analysis of Real and Nominal Bracket Slot Heights"

_applsci, doi:10.3390/app121910120_

Round 1

Reviewer 1 Report

Introduction

1.       Some sentences require grammatical correction for instance: “3D printing is being adopted at an increasing rate in various fields of dentistry, among them [is] Orthodontics”.

2.       The authors mentioned that CAD-CAM is superior to ceramic brackets in many aspects. Perhaps, the authors could list down a few of these aspects.

Materials and Methods

1.       The sample description in the materials and methods section is slightly different from the abstract. Perhaps, the authors would like to rephrase the abstract version as it is assumed that only 10 brackets were used in the study based on the abstract.

2.       The manuscript stated tooth 21 but Table 1 shows tooth 11?

3.       Perhaps the authors would like to change from ‘Group 4. In-house 3D-printed brackets (IH3DB):’ to ‘Group 4. In-house 3D-printed [resin] brackets (IH3DB)’

4.       I do understand that the authors are trying to compare different types of brackets (5 commercialized and 2 in-house printed) as well as categorized them into 4 different groups based on their materials. However, please be mindful that such a complex grouping may lead to confusion among readers. Perhaps, the authors would like to add on another hypothesis as the objective stated is only ‘null hypothesis was that there would be no difference in this regard between 90 commercially available brackets and those 3D-printed in-house’ which we presume is among the 7 types of brackets instead of the 4 grouping of materials.

5.       Please remove the sub-heading ‘method’

6.       Figure 1 to Figure 4 can be merged into one figure. Figure 5 can be a stand-alone figure.

7.       Finally, within Groups 1, 2 and 4, each characterized by the presence of two types of brackets, the degrees of slot height accuracy of each brand were compared using…” However, from the table it can be seen that Group 3 is also part of the analysis.

Results

1.       What is SD? Please write the full name and the word ‘SD’ can be used for the subsequent sentences.

2.       Please summarize Table 3 by including important data only such as the sample, mean, SD, SE and merge the p value in table 4 into table 3. There are too many tables in the manuscript.

3.       Table 6 can be merged below Table 5.

Discussion

1.       Remove Figure 6 and add one paragraph of [limitations and future recommendation].

Author Response

Dear Reviewer and Editor,

Below a point by point where each criticism raised is discussed.

Each modification in the main file is red marked.

We hope you appreciate our effort to strengthen the quality of our manuscript.

Reviewer 1

Introduction

  1. Some sentences require grammatical correction for instance: “3D printing is being adopted at an increasing rate in various fields of dentistry, among them [is]Orthodontics”.

We have performed grammatical correction of the manuscript you required.

  1. The authors mentioned that CAD-CAM is superior to ceramic brackets in many aspects. Perhaps, the authors could list down a few of these aspects.

Thanks for the comment, we have already written some possible advantages regarding the production of CAD/CAM brackets, such as customization according to clinical needs and its cost-effectiveness for both clinician and patient.

Materials and Methods

  1. The sample description in the materials and methods section is slightly different from the abstract. Perhaps, the authors would like to rephrase the abstract version as it is assumed that only 10 brackets were used in the study based on the abstract.

We have modified the materials and methods section of the abstract

  1. The manuscript stated tooth 21 but Table 1 shows tooth 11?

Sorry for the mistake, we have changed the table to the correct number of the corresponding bracket

  1. Perhaps the authors would like to change from ‘Group 4. In-house 3D-printed brackets (IH3DB):’ to ‘Group 4. In-house 3D-printed [resin]brackets (IH3DB)’

Thanks for the tip, we've changed from In-house 3D-printed brackets to In-house 3D-printed resin brackets

  1. I do understand that the authors are trying to compare different types of brackets (5 commercialized and 2 in-house printed) as well as categorized them into 4 different groups based on their materials. However, please be mindful that such a complex grouping may lead to confusion among readers. Perhaps, the authors would like to add on another hypothesis as the objective stated is only ‘null hypothesis was that there would be no difference in this regard between 90 commercially available brackets and those 3D-printed in-house’ which we presume is among the 7 types of brackets instead of the 4 grouping of materials.

Dear reviewer, as already written in our objectives “the null hypothesis was that there would be no difference in this regard between commercially available brackets and those 3D-printed in-house”

  1. Please remove the sub-heading ‘method’

Thanks, We have removed as you suggested the sub-heading ‘method’

  1. Figure 1 to Figure 4 can be merged into one figure. Figure 5 can be a stand-alone figure.

Thanks for the tip, we have grouped photos 2, 3 and 4

  1. Finally, within Groups 1, 2 and 4, each characterized by the presence of two types of brackets, the degrees of slot height accuracy of each brand were compared using…” However, from the table it can be seen that Group 3 is also part of the analysis.

Dear reviewer, Groups 1, 2 and 4 were analyzed with Wilcoxon's non-parametric test (p <0.05) (table 5) because they are the only ones to have two types of brackets in order to reveal any differences attributable to different production methods was used.

Results

  1. What is SD? Please write the full name and the word ‘SD’ can be used for the subsequent sentences.

Dear Reviewer, as already written in the article, SD indicates standard deviation. "Descriptive statistics were produced for each group, reporting the respective mean and standard deviation (SD)."

  1. Please summarize Table 3 by including important data only such as the sample, mean, SD, SE and merge the p value in table 4 into table 3. There are too many tables in the manuscript.

Thank you for your suggestion. We have reduced the number of tables and merged table 3 with table 4

  1. Table 6 can be merged below Table 5.

Dear reviewer, Table 4 compares the 4 macro groups with each other, while Table 5 compares groups 1,2 and 4 to evaluate to reveal any differences attributable to different production methods was used

Discussion

  1. Remove Figure 6 and add one paragraph of [limitations and future recommendation].

Thanks for the advice, we have removed figure 6 and added a paragraph on limitations and future recommendation

Reviewer 2 Report

The article is well structured and the research is conducted in accordance with scientific requirements. I appreciated the idea of the authors,  it is an interesting study and I think it can be a starting point for other research in this direction. I suggest the authors to apply and verify the results in vivo.

I have some questions/recommendations to clarify some of the authors' less-emphasized points.

1.      Tables and figures are not found in the article. The tables, figures and explanation are not found in the article. I only found the legend at the end of the article.All Figures, Schemes and Tables should be inserted into the main text close to their first citation and must be numbered following their number of appearance (Figure 1, Scheme I, Figure 2, Scheme II, Table 1, etc.).All Figures, Schemes and Tables should have a short explanatory title and caption.

2.      I did not find the limits of the study specified

3.       The references chapter does not fully respect the recommended style

4.     Do you think that the obtained results are also valid in vivo?

5.     Are you thinking of continuing your research and if so, in what way?

Author Response

Reviewer 

The article is well structured and the research is conducted in accordance with scientific requirements. I appreciated the idea of the authors, it is an interesting study and I think it can be a starting point for other research in this direction. I suggest the authors to apply and verify the results in vivo.

Thank you for your compliments. We have already started clinical trials, so far with excellent results

I have some questions/recommendations to clarify some of the authors' less-emphasized points.

  1. Tables and figures are not found in the article.The tables, figures and explanation are not found in the article. I only found the legend at the end of the article.All Figures, Schemes and Tables should be inserted into the main text close to their first citation and must be numbered following their number of appearance (Figure 1, Scheme I, Figure 2, Scheme II, Table 1, etc.).All Figures, Schemes and Tables should have a short explanatory title and caption.

We have added figures and tables within the main manuscript

  1. I did not find the limits of the study specified

We have added a paragraph with limitations and future recommendation

  1. The references chapter does not fully respect the recommended style

We have corrected the style of references

  1. Do you think that the obtained results are also valid in vivo?

We think that the results obtained are also valid in vivo, clinically we will have a reduction of the play wire-slot that will lead to a variation of the current prescriptions, created to contrast the current play wire-slot. The clinical tests performed so far show a complete expression of the prescriptions

  1. Are you thinking of continuing your research and if so, in what way?

We are continuing our research by performing in vitro tests to evaluate friction resistance, resistance to torque loads, FEM simulations to evaluate various clinical conditions and clinical tests solving, at this time, simple malocusions.

Reviewer 3 Report

Dear authors, thank you for the submission. The manuscript is well-written and well-structured, however, there is one issue that needs to be solved.

In the main text there are citations of figures and tables, as well as legends of figures and tables, however, the corresponding figures and tables are absent in the current attached article.

The absence of figures and table compromise an adequate review.

Author Response

Reviewer 3

Dear authors, thank you for the submission. The manuscript is well-written and well-structured, however, there is one issue that needs to be solved.

In the main text there are citations of figures and tables, as well as legends of figures and tables, however, the corresponding figures and tables are absent in the current attached article.

The absence of figures and table compromise an adequate review.

Thanks for the comment, sorry for the mistake. We have inserted tables and figures in the main text

Round 2

Reviewer 1 Report

The authors have clearly revised the manuscript. No further amendment is needed. 

Author Response

Dear Reviewer and editor,
Thank you for appreciating our manuscript

Reviewer 2 Report

In my opinion, the article complies with the publication requirements.

Author Response

Dear Reviewer and Publisher,
Thank you for appreciating our manuscript

Reviewer 3 Report

Dear authors, thank you for resubmitting the manuscript with appropriated figures and tables. Some point need to be improved:

“The purpose of this study was therefore to investigate the slot height accuracy of in-house 3D-printed resin brackets, as compared to other types of brackets on the market.” Be specific of the other types of brackets used in your study.

In figure 1, the brackets were digitally designed and 3D printed, however, all these brackets are fixed on a “platform”, to allow the preparation. How were these brackets separated or removed? Can these actions interfere with the slot height?

“The reproducibility of each manual measurement performed was good, as the p value was always ˃0.05 (Table 2)” What did you mean by good? Is there any classification to characterize the p-values as good or not?

Figure 3: there are some words in Italian. Consider translating into English, as present in the manuscript.

Figure 3: “The range of measurements made was expressed as mean and SD range” Unfortunately, the mean and standard deviation are not represented in the figure. As well as the statistical representations that indicate the differences are not represented in the figure.

As performed non-parametric analysis, the median and maximum / minimum values are more interesting to be presented.

Tables 3-5 could be reduced and better structured. Some ‘spaces’ are cut off making it impossible to see the text.

Hypothesis: I recommend that the study hypothesis be considered and discussed in the discussion topic, rather than the conclusion.

Author Response

Dear Reviewer and Editor,

Below a point by point where each criticism raised is discussed.

Each modification in the main file is red marked.

We hope you appreciate our effort to strengthen the quality of our manuscript

Reviewer

Dear authors, thank you for resubmitting the manuscript with appropriated figures and tables. Some point need to be improved:

“The purpose of this study was therefore to investigate the slot height accuracy of in-house 3D-printed resin brackets, as compared to other types of brackets on the market.” Be specific of the other types of brackets used in your study.

We corrected the purpose of the study by adding all the types of brackets used

In figure 1, the brackets were digitally designed and 3D printed, however, all these brackets are fixed on a “platform”, to allow the preparation. How were these brackets separated or removed? Can these actions interfere with the slot height?

Thanks for the question, as in figure 1 during the position in the print bed the support pins have been positioned away from the slot thus allowing a safe detachment without compromising the height of the slot

“The reproducibility of each manual measurement performed was good, as the p value was always ˃0.05 (Table 2)” What did you mean by good? Is there any classification to characterize the p-values as good or not?

Measurements were conducted by a single operator (CC) and repeated two weeks apart. in the statistical analysis the values were p value> 0.05, confirming that our measurement method was found to be reliable. The statistician considered it correct to set the p-value to 0.05

Figure 3: there are some words in Italian. Consider translating into English, as present in the manuscript.

We corrected the words in Italian, sorry for the mistake

Figure 3: “The range of measurements made was expressed as mean and SD range” Unfortunately, the mean and standard deviation are not represented in the figure. As well as the statistical representations that indicate the differences are not represented in the figure.

As performed non-parametric analysis, the median and maximum / minimum values are more interesting to be presented.

We have revised table number 3, we have added the median, the other values such as SD, maximum and minimum and are already present in the table

Tables 3-5 could be reduced and better structured. Some ‘spaces’ are cut off making it impossible to see the text.

Hypothesis: I recommend that the study hypothesis be considered and discussed in the discussion topic, rather than the conclusion.

Thanks for the advice, we discussed our null hypothesis in the discussion, and corrected the tables by reducing the spaces

Round 3

Reviewer 3 Report

Dear author, I have no more questions. Thank you.